# Moral distress related to paid and unpaid care among healthcare workers during the COVID-19 pandemic

**Julia Smith**[1], **Muhammad Haaris Tiwana**[1]*, **Alice Murage**[1], **Hasina Samji**[1], **Rosemary Morgan**[2], **Jorge Andres Delgado-Ron**[1]

**1** Faculty of Health Sciences, Simon Fraser University, Burnaby, BC, Canada, **2** Department of International Health, Bloomberg School of Public Health, Johns Hopkins University, Baltimore, MD, United States of America

☯ These authors contributed equally to this work.
* mhtiwana@sfu.ca

## Abstract

While there is growing literature on experiences of healthcare workers and those providing unpaid care during COVID-19, little research considers the relationships between paid and unpaid care burdens and contributions. We administered a moral distress survey to healthcare workers in Canada, in 2022, collecting data on both paid and unpaid care. There were no significant differences in the proportion of participants providing unpaid care by gender, with both genders equally affected by certain responsibilities such as reduced contact with family/loved ones. However, men were significantly more distressed about specific unpaid care responsibilities. Unpaid care was not significantly associated with differences in intention to leave work. At work, women were significantly more concerned about patients unable to see family, while men were distressed by others mistreating COVID patients. This study enhances understanding of paid and unpaid care relationships, particularly during crises, and proposes an innovative method for assessing unpaid care burdens.

## Introduction

The COVID-19 pandemic caused a dual crisis in the care economy—demands on healthcare workers increased, as did demands on those providing unpaid caregiving within the home and community. The immediate effects of this dual crisis are well-documented [1–3]. Healthcare workers experienced a greater risk of COVID-19 infection and secondary health effects amidst the crisis, such as anxiety and depression [4]. Those providing unpaid care (both health and non-health related) also saw work burdens increase due to service interruptions and isolation requirements, thereby impairing their mental health, resulting in economic and employment loss, and reduced educational opportunities for their dependents [5–7]. These crises were gendered; women make up 70 percent of the healthcare workforce around the world and do two-to-three times more unpaid care work than men [8]. In several contexts, women healthcare workers were more likely to contract COVID-19, partly due to the type of occupations they are

**Data Availability Statement:** The de-identified dataset is available on the Open Science Framework (OSF) repository, available from the following: DOI 10.17605/OSF.IO/4VXGE.

**Funding:** This work was supported by the Canadian Institute for Health Research (CIHR) URL: https://cihr-irsc.gc.ca/e/193.html through grant number 483891 received by JS. The funder had no role in the design of the study; in the collection, analyses, or interpretation of data; in the writing of the manuscript, nor in the decision to publish the results.

**Competing interests:** The authors have declared that no competing interests exist.

most likely to fill in the health system, which involve direct and prolonged contact with patients [9].

While policy and scholarship often position unpaid care as independent from paid health-care provision—for example, by focusing on benefits and training for paid care workers and not family carers—feminist care economy research has shown that both forms of care depend on each other [10]. Unpaid care refers to rendering caregiving, support, or assistance to those within one's social sphere - including family, friends, and community members - without compensation [11]. Unpaid caregivers provide essential day-to-day support, assisting with tasks that might not be covered by formal healthcare systems, such as personal care, companionship, and basic medical tasks. This informal caregiving sector is estimated to equal around 9% of the global gross domestic product. For instance, in Canada, this equates to 5 billion dollars of paid labour every year [12, 13]. Other forms of care, such as child and elder care, provide essential labour for the health and development of others. Such provision enables other workers, including healthcare workers, to participate in paid labour. The crisis in the care economy caused by the response to the COVID-19 pandemic provided a critical opportunity to explore the relationship between paid and unpaid care under systemic constraints. However, little research considers both the increased workload of healthcare workers and the burden of unpaid care during the pandemic in tandem, and more so under a feminist lens.

The COVID-19 pandemic also reinvigorated research into moral distress among healthcare workers. Moral distress, a concept that developed first in the nursing literature, aims to elucidate how structural constraints impose ethical dilemmas on healthcare workers, with negative impacts not only on care provision but also on the wellbeing and retention of healthcare workers [14]. While there is some debate about definitions and approaches to moral distress, here we define it as "the psychological unease or distress that occurs when one witnesses, does things or fails to do things that contradict deeply held moral and ethical beliefs and expectations."" [15]. High levels of moral distress are associated with poor mental and physical health, as well as burnout and attrition [16, 17]. Research on moral distress during the first three years of the COVID-19 pandemic documented sources of moral distress related to the shift from patient-centred to public health-focused approaches, a lack of resources that limited workers' capacity to provide high-quality care, and negative effect of enforced restrictions on the perceived quality of care [18, 19]. These sources of moral distress are associated with increased stress, anxiety, sleep disturbances, feelings of helplessness, and levels of empathy among healthcare workers [20]. While these outcomes may have multiple influencing factors, the inability to act ethically has been associated with impaired mental health, burnout, and attrition. Consequently, understanding the significance and contributors to moral distress is vital not only for supporting the well-being of care providers, but also for the overall quality and effectiveness of care delivery.

Despite numerous studies on moral distress among healthcare workers during the COVID-19 pandemic, and literature on unpaid care burdens during the pandemic, there appears to be little research on the intersection between moral distress and unpaid care work. In a qualitative research study with 88 women healthcare workers conducted in British Columbia (B.C.), Canada between December 2020 and March 2021, Smith and colleagues [21] found multiple and complex expressions of moral distress in response to paid and unpaid care during the COVID-19 pandemic. Participants expressed moral distress not only in response to their professional obligations, but also when reflecting on unpaid caregiving, such as child and elder care, and due to the conflict of managing both professional and personal care responsibilities during the pandemic [21]. Select studies have found a relationship between being a parent or having dependents, and professional moral distress, but do not explore this relationship [22]. To our knowledge, there are no previous quantitative studies specifically on moral distress related to

unpaid caregiving. Consequently, while there are numerous validated scales for assessing degree and sources of moral distress related to paid care work, there are no standard scales for measuring moral distress experiences related to unpaid care, likely due to a heightened focus on the workplace as a place of systemic constraints.

Recognizing that gender roles and social policies also create systemic constraints on unpaid care, this paper seeks to explore the concept of moral distress as applied to both paid and unpaid caregiving during the first two years of the COVID-19 pandemic. We apply the concept of moral distress, drawing on survey data with 2349 healthcare workers (physicians, nurses, homecare aids, and long-term care aids) in B.C., Canada, to examine how healthcare workers were affected by changes in both paid and unpaid care work, and explore the relationship between these experiences. We analyze our results by gender, an approach limited in the extant literature, as shown by a recent scoping review [23]. We specifically analyze gender differences in degree and source of moral distress related to both unpaid and paid care and explore the relationships between these. Through this research, we aim to not only advance research on moral distress in terms of gender analysis and a focus on unpaid care, but also research aimed at documenting the scale and effects of unpaid care burdens. Findings offer insights to guide gender sensitive policies that support care providers at work and home, during and following health crises.

## Context

This study was based in B.C., Canada where approximately 375,400 cases and 4,806 COVID-19-related deaths were recorded by the end of 2022 (when we completed data collection) [24, 25]. The majority outbreaks and deaths occurred in health or long-term care (LTC) facilities [26]. Pre-COVID-19, many healthcare workers already worked in environments where external constraints—related to public funding restrictions and the marketization of care over the past three decades—restricted their ability to provide quality care [27, 28]. Social policy in Canada has also failed to keep up with demand for paid childcare and eldercare, resulting in continued reliance on unpaid caregivers [29–31]. In this context, women remain the primary care providers within households and families, doing two to three times more unpaid care work than men, as gender norms continue to position unpaid care as women's responsibility [8]. During the first year of the COVID-19 pandemic, unpaid care responsibilities forced a disproportionate number of women, compared to men, out of paid work as they filled gaps caused by school, childcare, and service interruptions. For example, women made up over 90% of those who left work in the health and social care sector, in Canada, during the first three months of the pandemic [32]. Subsequent temporary closures and isolation periods continued to impose heightened unpaid care burdens throughout the pandemic. Attesting to the fact that women were more likely to take on the bulk of care responsibilities for dependents created by schooling and service interruptions [7], affecting their mental well-being, leading to economic setbacks, job losses, and diminished educational prospects for those they supported [5, 6]. As this context is not unique to high-income liberal democracies that experienced ongoing waves of COVID-19 infection, and implemented moderate containment strategies, findings from this case study may point to similar trends elsewhere, though case study specific research is required to understand contextual differences.

## Methods

### Positionality

We come to this topic as a diverse group of researchers with our own past experiences providing paid and unpaid caregiving, as well as of the COVID-19 pandemic. We recognize this

positionality influences our analysis, particularly in terms of the power we hold as researchers analyzing data that others have generously provided. Though we are a diverse group—in terms of gender, age, race, and ethnicity—we are all privileged in terms of educational attainment and employment. As such, we are at risk of the privilege hazard of not being best positioned to understand the challenges experienced or solutions to them [33]. We have aimed to mitigate this by sharing preliminary research findings with participants to ensure our analysis aligns with their experiences, inviting feedback, and contextualizing findings within the broader context of gender inequities in Canada.

## Sample and data collection

Details about recruitment have been described elsewhere [34]. The study was approved by the Simon Fraser University (SFU) Research Ethics Board (#30001218). In short, we administered an online questionnaire via 'SurveyMonkey' to a convenience sample of physicians, nurses, home and community care providers, and long-term care providers working in B.C., Canada [34, 35]. We recruited through email and social media via work unions, such as the Hospital Employees Union, and occupational organizations, such as SafeCare BC. Prior to participation, written consent was obtained from all participants at the onset of the survey. Only those who provided consent were eligible to proceed with completing the survey. We collected socio-demographic data and information about experiences of COVID-specific moral distress and unpaid care work moral distress during the last quarter of 2022, specifically between 3rd October and 30th December 2022, asking participants to reflect on experiences over the past year (since October 2021). Among the 2,918 healthcare workers who provided full outcome data, 2,349 answered the question, '*Do you provide care to relatives or friends (children, seniors, those with disability or chronic illness) at home or in your community (i.e., outside of work)? Yes/No.*' Given the study's focus on moral distress emanating from both paid and unpaid care work, the main analyses were conducted on this subset of the participants performing care at the workplace (paid care) and at home or the community (unpaid care). Respondents were allowed to skip questions to mitigate forced recollection of uncomfortable or distressing experiences.

## Item development: Moral distress at the workplace (paid) and specific to unpaid care in the home or community

Items for the survey were developed by bringing together findings from previous qualitative research and a literature review, together with existing moral distress survey items. From 2020 to 2022, some of the authors conducted qualitative research with healthcare workers in BC, through which they identified themes related to moral distress in response to paid and unpaid care [21]. We then conducted a scoping review of moral distress literature that included gender analysis and/or unpaid care (we found no papers on unpaid care) [23]. With findings from this previous research as well as the objectives of the study in mind, we reviewed existing moral distress surveys to identify items that could be adopted or adapted to answer our research questions related to moral distress during COVID-19 and in response to unpaid care. For example, we adopted seven items from the COVID-19 specific scale developed by Cramer & colleagues that also corresponded to findings from our previous qualitative research [36]. As we found no existing tools related to moral distress in response to unpaid care, items from the revised Moral Distress Scale (MDS-R) that corresponded with themes from previous qualitative research were adopted [37]. For example, as previous research on unpaid care included expressions of distress in response to having to care for children or older adults with special needs when home or school care was not available [21], we adapted the MDS-R question "Be required to care for patients I don't feel qualified to care for," to "I was required to care for

dependents whom I do not feel qualified to care for". Similarly, "Watch patient care suffer because of lack of provider continuity" became "Witnessed my dependents' health suffer due to inadequate care from professional providers or interruptions in services." In selecting questions, we were mindful of the length of the survey and so only selected those directly related to studies objectives. This was a dialectic process that included multiple discussion among the project team, revisiting previous research and reviewing multiple tools. Through this process we developed two sets of items for this study: unpaid caregiving at home/community and caregiving at the workplace, both occurring during the COVID-19 pandemic.

The scales resembled the structure of the original moral distress scale, with two components for each item: frequency (valued 0 to 4) and intensity (0 to 4). When multiplied, the components generate an item score (0 to 16). The sum of the seven items on each scale constitutes the total score for moral distress (0 to 112). The total scores, while related, were kept as two separate constructs for analytical purposes. Both questionnaires were piloted with nine respondents to assess readability, including some whose first language was not English.

## Descriptive statistics

We estimated counts and proportions of those who did and did not provide care outside of work to identify significant differences among these two groups by: age (Under 30, 30–39, 40–49, and 50+), gender (men and women), race (Indigenous, White, and Persons of Colour), profession (physicians, nurses, home and community care providers, and long-term care providers), intention to leave work (no, considered leaving, left), and the mean level of moral distress. We define race as a social construct based on historical and ongoing ideological categorization based of individual physical characteristics, most notably skin colour. This construct has real consequences on lived experiences due to systemic racism (differential distribution of public resources) and interpersonal racism. In consideration of racialization in the Canadian context, we include in our analysis three racial categories: Indigenous people, White people, and Persons of Colour. The latter represent people who self-identified as non-White. Gender is often conflated with biological sex. We define gender as socially constructed roles, behaviours, and attributes. In the survey, participants could choose one or more of the following gender-related identities: woman, man, two-spirit, transgender, non-binary, man, and other. In this analysis, we utilize data of those who identified either as woman or man due to sample limitation. We used the chi-square test and the Bonferroni test for multiple comparisons with an alpha level of 0.05 for categorical variables and the ANOVA for moral distress scores.

## Gender differences in moral distress at the workplace (paid) and (unpaid) at home/community

We created box plots to explore the variance of independent components of moral distress in response to care responsibilities at the workplace (paid) and in the home or community (unpaid). We examined differences in moral distress across groups through permutation tests to accommodate for potential skewness or outliers in our data. We utilized both mean and median as central tendency measures and 10,000 permutations per variable. Differences were considered significant if either measure of central tendency was below 0.05. Because we anticipated that gender influenced potential confounding effects of profession and age, we fitted an ANCOVA model that included these variables plus gender as predictors and the total score for items as the outcome. Subsequently, we computed the marginal mean scores and corresponding 95% confidence intervals, using the 'emmeans' R package, to obtain unbiased gender estimates of the total workplace (paid) and home/community (unpaid) care score.

### Exploratory factor analysis (EFA)

The EFA was used to explore the relationship between the two constructs represented by the total scores of home/community care moral distress and workplace moral distress. We used the 'psych' R package to run an EFA. First, we tested the sample adequacy through the Kaiser-Meyer-Olkin (KMO) measure and Bartlett's test of sphericity. Values above 0.6 and below 0.05, respectively, were deemed appropriate. A correlation matrix was created to explore collinearity. Then, we conducted a parallel analysis to determine the number of factors to retain in the EFA with oblique ('oblimin') rotation. Factor loadings were prespecified with 0.3 (minimum) and 0.4 (moderate) thresholds. The latter was used to assign constructs to each factor.

## Results

### Correlates of home and community unpaid care among healthcare workers

The prevalence of unpaid care in the sample was 43.8% (n = 1028). We did not find significant differences in the provision of unpaid care (yes vs. no) by gender or age. The proportion of home and community care providers (10.0%) was slightly higher among those who performed unpaid care compared to those who did not (6.9%; p-value = 0.05). Similarly, the proportion of non-Indigenous participants identifying as Person of Colour (POC) among unpaid care providers was significantly higher compared to the group that did not care for others without compensation (34.7% vs. 20.7%; p < 0.001). Conversely, the proportions of Indigenous (9% vs. 14.7%) and White participants (56.3% vs. 64.5%) providing unpaid care was significantly lower (p < 0.001 for both comparisons). Those who provided unpaid care had significantly higher mean moral distress levels related to paid care work (37.87 ± 24.49 SD) compared to those who were solely paid carers (34.72 ± 24.25 SD; p = 0.002). Overall, unpaid care was not significantly associated with differences in intention to leave work.

### Gender differences related to home and community unpaid care

Table 1 shows the characteristics of workers who performed both paid and unpaid care, stratified by gender. These subsamples were heterogeneous in terms of age, race, and profession. While the mean levels of moral distress at the workplace were higher for women, the mean score for unpaid care distress was higher for men (Table 1). Once the mean score was adjusted by profession and age, women's unpaid care distress score (41.5; 95% CI: 38.7 to 44.3) remained lower than that of men (45.5; 95% CI: 41.2 to 49.8), but the difference was no longer statistically significant.

In terms of sources of moral distress, the component analysis showed that both genders were equally affected by having reduced contact with family and loved ones to reduce the risk of COVID transmission (the highest burden for both groups), and by the lack of time to provide the physical and emotional care and educational or other developmental support their dependents needed. Men were significantly more distressed when (a) they were required to care for dependents whom they did not feel qualified to care for, (b) their obligations to care for dependent(s) with COVID transmission risk, indirectly extended the risk of transmission to patients/residents, and when they (c) witnessed their dependents' health, (d) education, or overall development suffer due to inadequate care from professional providers or interruptions in services, including school (Fig 1).

### Gender differences related to moral distress at the workplace

Compared to men, women were significantly more concerned about caring for patients/residents who could not see their family or friends while in the facility (their main concern),

**Table 1. Characteristics of workers who provided caregiving both in the workplace (paid) and at home or in their community (unpaid), stratified by gender.**

|  | Men | Women |
|---|---|---|
|  | n = 144 | n = 689 |
| Profession (%) |  |  |
| Home and Community care provider | 44 (30.6) | 42 (6.1) |
| Long-term care provider | 44 (30.6) | 511 (74.2) |
| Nurse | 6 (4.2) | 102 (14.8) |
| Physician | 50 (34.7) | 34 (4.9) |
| Age (%) |  |  |
| Under 30 | 14 (9.7) | 164 (23.9) |
| 30–39 | 64 (44.4) | 214 (31.1) |
| 40–49 | 58 (40.3) | 182 (26.5) |
| 50+ | 8 (5.6) | 127 (18.5) |
| POC (%) |  |  |
| Indigenous | 30 (20.8) | 47 (6.9) |
| POC | 17 (11.8) | 170 (24.9) |
| White | 97 (67.4) | 467 (68.3) |
| Unpaid care score (mean (S.D.)) | 44.94 (25.60) | 38.60 (24.35) |
| Workplace distress score (mean (S.D.)) | 38.60 (21.91) | 41.96 (24.84) |

POC: Person of colour; SD: Standard deviation.

followed by caring for patients who presented transmission risk to their dependents. The latter was the main concern for men, although women had significantly higher mean scores with no significant differences in median scores. Men were significantly more distressed by working

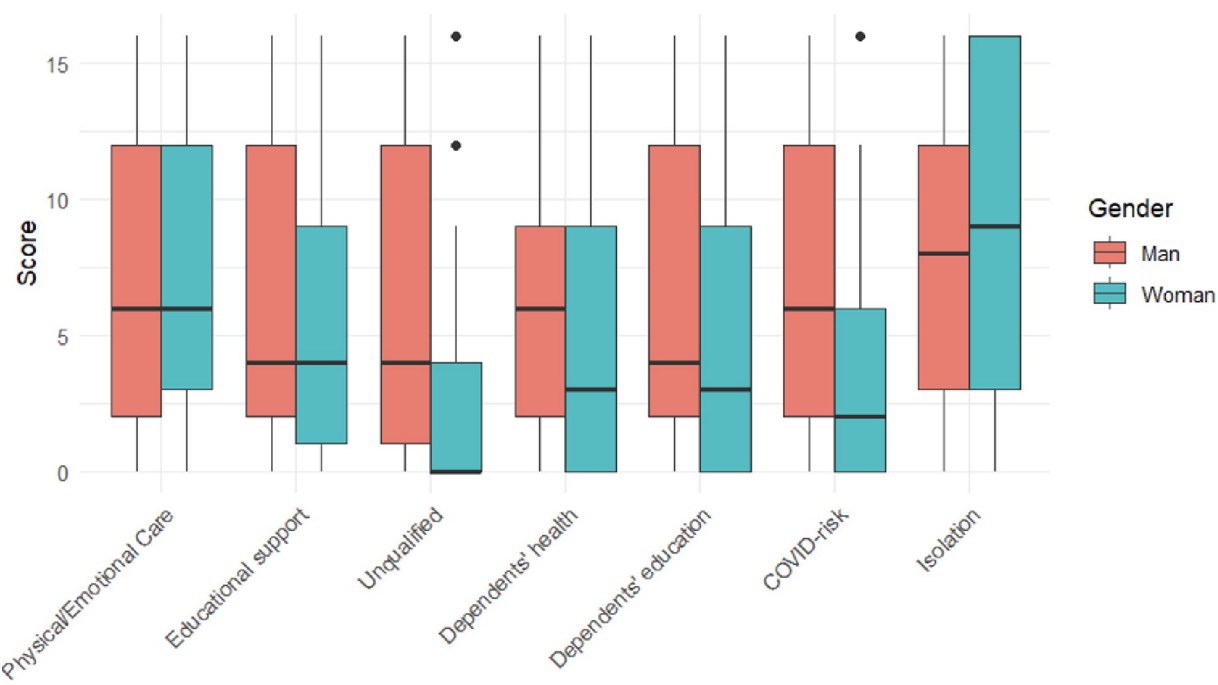

**Fig 1. Box plots of items for unpaid care distress at home or the community by gender.**

with team members who did not treat those with COVID with dignity and respect. Both genders were equally distressed by the lack of personal protective equipment (PPE) or cleaning supplies to avoid COVID transmission at work, the pressure to order or carry out COVID-related protocols that they considered unnecessary or inappropriate, and to overemphasize COVID related policies at the expense of patient care. Some women were more distressed by caring for patients/residents who died while in hospital/care without family, friends and/or clergy present, which is reflected in higher means but no difference in median scores (Fig 2).

### Exploratory factor analysis

The KMO values ranged from 0.84 to 0.92, with an overall value of 0.88. The p-value for the Barlett test was <0.001 and the determinant of the correlation matrix was 0.006. Therefore, EFA was deemed appropriate. The parallel analysis resulted in 5 factors. Table 2 shows the results of the EFA:

Most factors loaded independently in subgroups that belonged to the original groupings; factor 1 ("lack of time"), 4 ("transmission concerns") and 5 ("feeling unqualified") for unpaid care items and factors 2 ("Lack of PPE and patient isolation") and 3 ("inappropriate protocols") for workplace items. Caring for patients who presented transmission risk to the healthcare workers' dependents cross-loaded on factors 2 and 4, whereas witnessing the impact of inadequate professional care or the interruption of services on their dependent's health cross-loaded on factors 1 and 4, suggesting an overlap of constructs for this item as a source of moral distress both at home and at the workplace.

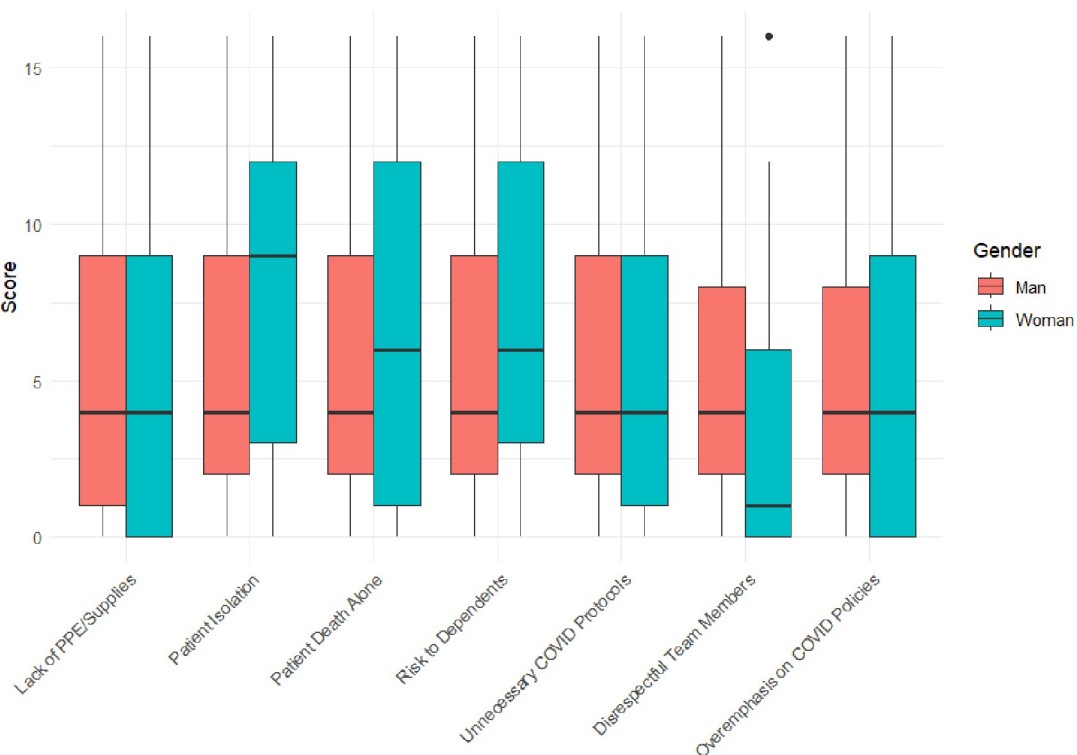

**Fig 2. Box plots of items for COVID-specific distress at the workplace by gender.**

**Table 2. Results of exploratory factor analysis.**

| Original group | Item | Factor 1 | Factor 2 | Factor 3 | Factor 4 | Factor 5 |
|---|---|---|---|---|---|---|
| PAID CARE | Have not had access to PPE or cleaning supplies I feel necessary to avoid COVID transmission at work | 0.14 | **0.53** | 0.06 | -0.15 | 0.31 |
| PAID CARE | Cared for patients/residents who are prevented from seeing family or friends while in the facility | 0.05 | **0.69** | 0.07 | 0.03 | -0.15 |
| PAID CARE | Cared for patients/residents who die while in hospital/care without family, friends and/or clergy present. | -0.01 | **0.77** | 0.00 | 0.03 | 0.07 |
| PAID CARE | Cared for patients who present transmission risk to my dependents. | 0.08 | 0.35 | 0.22 | 0.30 | -0.19 |
| PAID CARE | Felt pressured to order or carry out COVID-related protocols that I consider to be unnecessary or inappropriate. | 0.07 | 0.02 | **0.72** | -0.07 | -0.05 |
| PAID CARE | Worked with team members who do not treat those with COVID with dignity and respect. | -0.16 | 0.19 | 0.33 | 0.13 | 0.28 |
| PAID CARE | Felt required to overemphasize COVID related policies at the expense of patient care. | -0.03 | 0.02 | **0.78** | 0.02 | 0.02 |
| UNPAID CARE | Lacked the time to provide the physical and emotional care my dependents need | **0.82** | 0.04 | 0.00 | 0.03 | -0.08 |
| UNPAID CARE | Lacked the time to provide the educational or other developmental support my dependents need | **0.75** | 0.02 | 0.04 | -0.03 | 0.14 |
| UNPAID CARE | Was required to care for dependents whom I do not feel qualified to care for. | 0.12 | 0.04 | 0.07 | 0.17 | **0.60** |
| UNPAID CARE | Witnessed my dependents' health suffer due to inadequate care from professional providers or interruptions in services | 0.31 | -0.09 | 0.14 | 0.35 | 0.12 |
| UNPAID CARE | Witnessed my dependents education or overall development suffer due to interruptions in schooling/services. | 0.29 | -0.11 | 0.29 | 0.26 | 0.08 |
| UNPAID CARE | Cared for dependent(s) with COVID transmission risk, indirectly extending transmission risk to patients/residents | 0.00 | 0.01 | 0.09 | **0.59** | 0.26 |
| UNPAID CARE | Reduced contact with my family and loved ones to reduce risk of transmission of COVID | 0.25 | 0.22 | -0.07 | **0.46** | -0.15 |
| | Sum of squares | 1.87 | 1.80 | 1.73 | 1.23 | 0.85 |
| | Proportion of variance | 13% | 13% | 12% | 9% | 6% |
| | Cumulative variance | 13% | 26% | 38% | 47% | 53% |

Tucker Lewis Index of factoring reliability = 0.958. RMSEA index = 0. BIC = -111.5

## Discussion

To our knowledge, there is limited/sparse literature on moral distress during COVID-19 that considered gender [34, 35]. Our analysis demonstrated that men had higher moral distress scores related to unpaid care than women. Men were significantly more distressed when they were required to care for dependents whom they did not feel qualified to care for and witnessed their dependents' health, and education, or overall development suffer due to inadequate care from professional providers or interruptions in services, including school. These findings suggest that when men were asked to take on tasks that are often conducted by women (in addition to healthcare, teaching and childcare are highly feminized professions in Canada) they felt unqualified. This may reflect gendered assumptions about unpaid care that position care as something that all women can 'naturally' do, and by extension, not something men 'naturally' excel at, leading to their perception of feeling unqualified [38]. Furthermore, distress related to education and development may reflect evidence that home schooling was one of the forms of unpaid care that men were more likely to participate in during the pandemic, due to gender norms, and so were likely to be aware of related challenges [39]. Research has documented how traditional forms of masculinity conflict with increased demand on men to taking on caring roles, and how men are negotiating these tensions [40]. Hrženjak, among other care theorists, calls for a de-gendering of care as a means to both free women from

demands to provide care and create space for men to take on more caregiving [41]. Creating awareness—through research, policy and public discussion—around women's disproportionate contributions to unpaid care, while also providing paid care, can expose the contradictions and injustices imbedded in gender norms.

Women reported slightly higher moral distress scores at the workplace during the COVID-19 pandemic, with results highlighting that they were significantly more concerned about caring for patients/residents who could not see their family or friends while in the facility (their main concern), followed by caring for patients who presented transmission risk to their dependents. One possible explanation is the gendered nature of emotional labour in healthcare work; as previous research indicates, women are often expected to bear greater emotional labour burdens in the workplace compared to men [42]. In addition, other studies examining the gendered nature of the healthcare workforce note that women are more likely to be in positions with prolonged and close contact with patients, compared to men, leading to potentially greater opportunities for moral distress [43]. COVID-19 related sources of moral distress identified in our study, such as PPE, support existing literature as well [44]. The response to COVID-19 may have had similar, or potentially exacerbated, impacts on these gendered positionalities.

In previous qualitative research, healthcare workers expressed frustration that their identities as both paid and unpaid care providers are not recognized in the workplace, resulting in the absence of supportive policies in general and during COVID-19 [21], this may partly explain the significantly higher mean moral distress levels at the workplace among who provided unpaid care, compared to those who did not. It is perhaps not surprising that those central to the COVID response, in both public and private spheres, bore a greater moral distress burden (than those solely providing care at work), yet the relationship between paid and unpaid care experiences is rarely discussed. Apart from adding quantitative evidence to demonstrate particular moral distress burdens on those experiencing double care responsibilities, we explored relationships between structural factors determining moral distress in public and private spheres. We observed some intersections between these domains. Notably, when crises impact the health of dependents, this moral distress manifests in both professional care responsibilities and distress associated with unpaid caregiving.

In applying the concept of moral distress to unpaid, as well as paid care, we hope to introduce a further method of analyzing the costs and effects of unpaid care on those responsible for providing it. Currently, such assessments rely on time-use surveys and financial assessments—often using hourly earnings estimates to determine financial and opportunity costs [45]. While understanding the economics of unpaid care is crucial, such approaches do not capture psychosocial, emotional, and mental health burdens despite unpaid caregivers' past reports of related stress, anxiety, and depression [2]. During crises like the COVID-19 pandemic, these costs increase, with over 70% of women in one Canadian survey reporting mental health burdens related to increased unpaid care work during the first few months of the pandemic [46]. A later survey conducted in April 2021 found that 71% of mothers were "at the breaking point" due to stress and anxiety [47]. Documenting moral distress related to unpaid care offers a further approach to assessing the multidimensional burdens born by those providing care. In turn, facilitating the development of strategies to mitigate such burdens and their effects.

Due to the scope of our research project, we have specifically focused on unpaid care burdens of healthcare workers during the COVID-19 pandemic. However, future research might build off this example to develop a moral distress scale related to unpaid caregiving for the general population and in broader settings not specific to the COVID-19 crisis. Such research may provide an innovative approach to understanding the complex experience of providing unpaid

care and the psychosocial effects of such burdens. While moral distress often corresponds to burnout and attrition among healthcare providers [16], questions emerged about its long-term effects on unpaid care providers, who often do not have the option of forgoing care responsibilities.

This study has a number of limitations, including its cross-sectional nature and non-random sampling, which limit generalizability to the entire workforce. It may be that those healthcare workers most aware of, or concerned about moral distress, volunteered for the survey, influencing results. The timing of the study, following the emergency response phase when vaccines were available and health regulations eased, will have also influenced results, as will the recall bias of participants who were asked to reflect on experiences over the past year (since October 2021). While our main analyses were on gender, we do not examine intersecting identity factors related to race, ethnicity, ability, etc. Even within our analysis, the sample size restricted us to men and women genders, excluding those who identified as transgender, two-spirit, and non-binary. Specific research on these genders is needed to understand their experiences better, as is broader research on how intersectional inequities related to race, class etc. interact with experiences of moral distress. Such research might consider how health and care systems in countries such as Canada are not only distinctly gendered but also increasingly racialized and reflect notable socio-economic differences in terms of who fills what roles [23]. Similarly, a more nuanced assessment of unpaid care responsibilities, in terms of amount and type of care as well as in relation to different dependents (children versus older adults etc.), is required. Furthermore, we analyzed the unpaid care responsibilities using a single-item question, which may not capture the complexity and depth of this construct, such as type of care provision, amount of care etc. A restricted sample size might also have caused type I errors, where we could not find statistical differences that might be present at the population level. Hence, this finding should be interpreted with caution, as it may be less stable than findings based on more systematically measured constructs. Random sampling of healthcare workers and unpaid care providers would further provide more representative results of the general population.

## Conclusion

Our study has highlighted the complex dynamics and significant influence of gender and other potential identity factors in shaping moral distress among healthcare workers during a public health crisis. Findings that demonstrate greater levels of moral distress at work among those with unpaid care responsibilities, reinforce what healthcare workers have previously expressed [21]—their care work at home impacts their professional care work. Policies that reduce unpaid care burdens, such as accessible childcare and elder care, may mitigate moral distress, and related burnout. Considering the current crisis in human resources for health care in Canada and elsewhere such policies would not only benefit healthcare workers but the broader health system and therefore population level health. Employers may also take targeted actions such as enabling more flexible scheduling, improving caregiving leave policies, and offering support for care responsibilities (such as childcare subsidies, homecare assistance, etc).

Our findings, further emphasize the urgent need for gender-supportive approaches and research with a more comprehensive understanding of unpaid care burdens to better support those who primarily provide unpaid care in society. Findings related to men's elevated moral distress related to unpaid care indicate a need to counter gender norms that position unpaid care as women's expertise and to build men's confidence in providing unpaid care. Such change requires cultural shifts that while slow can be promoted through modeling of those in positions of authority and gender sensitivity trainings. Policies that support men's engagement in unpaid

care, such as shared parental leave and flexible work policies in masculine-dominated fields, can further promote change [48]. Such interventions might particularly be developed to support men healthcare workers through workplace trainings on caring masculinity [40].

There is a growing literature calling for greater recognition, support, and compensation for the emotional labour that women healthcare workers often provide [49, 50]. Findings here suggest such support is particularly crucial during a crisis and needs to include the emotional support that is provided to dependents. Indeed, feminist research on the effects of the COVID-19 pandemic on the care economy, clearly demonstrates the intrinsic relationship between unpaid and paid care, and the need to recognize that a strong healthcare workforce depends on supportive care policies, including accessible childcare, support with elder care and resources for selfcare, among others.

## Acknowledgments

We acknowledge and are grateful to Alexander Korzuchowski, Simran Purewal and Neda Zol-faghari who helped with preliminary research and survey development.

## Author Contributions

**Conceptualization:** Julia Smith, Alice Murage, Rosemary Morgan.

**Data curation:** Muhammad Haaris Tiwana.

**Formal analysis:** Muhammad Haaris Tiwana, Jorge Andres Delgado-Ron.

**Funding acquisition:** Julia Smith.

**Methodology:** Muhammad Haaris Tiwana, Jorge Andres Delgado-Ron.

**Project administration:** Muhammad Haaris Tiwana.

**Supervision:** Julia Smith.

**Writing – original draft:** Julia Smith, Muhammad Haaris Tiwana, Jorge Andres Delgado-Ron.

**Writing – review & editing:** Julia Smith, Muhammad Haaris Tiwana, Alice Murage, Hasina Samji, Rosemary Morgan, Jorge Andres Delgado-Ron.

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
