## [Decision Letter · Decision Letter 0]

25 Jun 2024

PONE-D-24-17880Moral distress related to paid and unpaid care  among healthcare workers  during the COVID-19 pandemicPLOS ONE

Dear Dr. Tiwana,

Thank you for submitting your manuscript to PLOS ONE. After careful consideration, we feel that it has merit but does not fully meet PLOS ONE’s publication criteria as it currently stands. Therefore, we invite you to submit a revised version of the manuscript that addresses the points raised during the review process.

Please review the following: 

I agree with the need to clarify the points detailed by the reviewers. Especially, detail the social relevance of the study, that is, how the research findings can be used and which people can use them. In this sense, indicate future lines of research precisely and in the context of recent scientific production.

We look forward to receiving your revised manuscript.

Kind regards,

Alejandro Botero Carvajal, MD

Academic Editor

PLOS ONE

2. PLOS requires an ORCID iD for the corresponding author in Editorial Manager on papers submitted after December 6th, 2016. Please ensure that you have an ORCID iD and that it is validated in Editorial Manager. To do this, go to ‘Update my Information’ (in the upper left-hand corner of the main menu), and click on the Fetch/Validate link next to the ORCID field. This will take you to the ORCID site and allow you to create a new iD or authenticate a pre-existing iD in Editorial Manager. Please see the following video for instructions on linking an ORCID iD to your Editorial Manager account: https://www.youtube.com/watch?v=_xcclfuvtxQ".

3. In the online submission form, you indicated that [The data will be made available upon reasonable request from first author.]. 

Reviewers' comments:

Reviewer's Responses to Questions

**Comments to the Author**

1. Is the manuscript technically sound, and do the data support the conclusions?

Reviewer #1: Yes

Reviewer #2: Yes

2. Has the statistical analysis been performed appropriately and rigorously? 

Reviewer #1: Yes

Reviewer #2: N/A

3. Have the authors made all data underlying the findings in their manuscript fully available?

Reviewer #1: Yes

Reviewer #2: Yes

4. Is the manuscript presented in an intelligible fashion and written in standard English?

Reviewer #1: Yes

Reviewer #2: Yes

5. Review Comments to the Author

Reviewer #1: The Introduction makes a reasonable and clear case for the relevance of this study and how it contributes to the current literature. I see the importance of this study as another needed step in broadening the field of moral distress from the inpatient setting.

The manuscript appropriately raises the challenge for the field of moral distress in ongoing debates on its definition (line 73) and on its overfocus on the moral distress and its measurement in hospital-based workers (97-100), specifically nurses. The authors took significant steps to develop survey questions appropriate for unpaid care in the home and community setting, and I generally like the items the authors choose and adapted (although any set group of measurement statements will inevitably exclude capturing moral distress from issues not included within the statements selected, a real risk for a phenomenon as broad as the causes of moral distress).

My issue is that some of the crafted items, e.g., “Witnessed my dependents’ health suffer due to inadequate care from professional providers or interruptions in services,” “Witnessed my dependents’ education or overall development suffer due to interruptions in schooling/services” and “Worked with team members who do not treat those with COVID with dignity and respect” do not fit the standard definition of moral distress as coming from constraints on an individual’s own actions and I believe reflected in the authors’ (perhaps too vague) conceptualization of moral distress (lines 71-72), “Moral distress can be conceptualized broadly as challenges faced by those unable to provide care they believe to be required.” Restated, by the survey measurement items selected, the authors’ definition of moral distress includes one’s distress from witnessing others’ actions or outside events that one feels are wrong, and not just distress stemming from actions taken when having one’s own actions constrained. It also includes distress from actions that are constrained from happening, and not just constraints forcing actions.

This is a definitional challenge for the entire field of moral distress and not unique to this study. I believe the authors do not disagree. The field needs to recognize that even some of its accepted measurement instruments take the concepts of moral distress beyond what stems from constraints on one’s own actions. If the authors see the field this way, here is one definition that acknowledges that moral distress stems from more than solely what an individual is forced to do themselves: “we conceptualise moral distress as the psychological unease or distress that occurs when one witnesses, does things or fails to do things that contradict deeply held moral and ethical beliefs and expectations.” (Pathman et al. Moral distress among clinicians working in U.S. safety net practices during the COVID-19 pandemic: A Mixed Methods Study. BMJ Open. 2022;12:e061369. doi: 10.1136/bmjopen-2022-061369). Clear and accurate definitions of a study’s fundamental concepts are, of course, crucial in science.

Specific comments:

1. Lines 22-23 I find the phrase of the abstract obtuse: “little research considers the interaction between these two sectors of the care economy.” Perhaps this can be reworded for clarity for those minimally familiar with the ins and outs of unpaid caregiving.

2. Line 119. Please clarify/explain/reword “extractive responses”. This sounds like jargon and perhaps too value laden for scientific writing.

3. Do the data allow investigators to know how both moral distress for both unpaid and paid care differ for subjects whose dependents include children? It would seem that the sources and amount of moral distress will differ when speaking of responsibilities and witnessing things happening to one’s own children versus other dependents. And given societal gender differences in responsibility for children (as the authors point out), how does parenthood affect moral distress differences in paid and unpaid care found for men and women?

4. Lines 350-362. The limitations listed are appropriate—I appreciate the investigators’ understanding of the limitations of their study and forthrightness in disclosing these limitations.

5. How are study findings specific to or potentially affected by the timing of the survey, i.e., at about 20 months into the pandemic, when PPE shortages had abated, and an effective vaccine was widely available?

6. Conclusion section. I find this section emphasizes study findings related to gender differences and omits conclusions from study findings about the amount and relationship between moral distress from unpaid and paid care, which is the manuscript’s title and the study’s seeming focus within the abstract.

Reviewer #2: I have carefully reviewed your manuscript titled "Moral distress related to paid and unpaid care among healthcare workers during the COVID-19 pandemic". I commend the authors for their insightful analysis of the gender-specific experiences of moral distress among healthcare workers during the COVID-19 pandemic, particularly focusing on unpaid caregiving responsibilities.

I commend you on the clarity and coherence of your manuscript, as well as the depth of your empirical analysis. Your policy recommendations are insightful and offer practical insights for policymakers and industry stakeholders. However, I recommend the following revisions to further strengthen your manuscript:

The manuscript effectively highlights the gender disparities in moral distress, the impact of gender norms on caregiving roles, and the interconnectedness of unpaid and paid care experiences. The discussion emphasizes the urgent need for gender-supportive approaches, comprehensive research, and policy interventions to address the complex dynamics of moral distress in healthcare settings.

Feedback and Recommendations:

Strengths Acknowledgement: The manuscript provides a comprehensive analysis of an important topic and offers valuable insights into the gendered experiences of moral distress among healthcare workers.

Revision Recommendation: Given the depth of the analysis and the importance of the topic, I recommend a major revision to strengthen the manuscript’s impact and relevance. Enhancing the discussion section with clearer implications for practice, additional policy recommendations, and a more detailed exploration of the intersectionality of gender and other identity factors would be beneficial.

Specific Suggestions for Revision:

1- In method: Clarify the rationale behind the selection of specific items for the moral distress scales and how they align with the study objectives and previous research findings.

2- Emphasize the practical implications of the findings for healthcare organizations, policymakers, and support systems for healthcare workers.

3- Address the study’s limitations, such as the non-random sampling and the exclusion of intersecting identity factors, and provide suggestions for future research directions to further explore these aspects.

Consider expanding on the recommendations for addressing gender norms, promoting men’s involvement in caregiving, and advocating for supportive care policies in healthcare settings.

4- In conclusion, with some revisions to enhance the practical implications and address study limitations, the manuscript has the potential to make a significant contribution to the field of healthcare ethics and gender studies. I look forward to seeing the revised manuscript and the valuable insights it will provide to the academic community.

6. PLOS authors have the option to publish the peer review history of their article (what does this mean?). If published, this will include your full peer review and any attached files.

Reviewer #1: **Yes: **Donald Pathman, MD MPH

Reviewer #2: **Yes: **Hadi Abbasian

---

## [Author Response · Author response to Decision Letter 0]

31 Jul 2024

Editorial comments

- I have updated the title page to reflect PLOS ONE's style requirements. Thank you for flagging.

2. PLOS requires an ORCID iD for the corresponding author in Editorial Manager on papers submitted after December 6th, 2016. Please ensure that you have an ORCID iD and that it is validated in Editorial Manager. To do this, go to ‘Update my Information’ (in the upper left-hand corner of the main menu), and click on the Fetch/Validate link next to the ORCID field. This will take you to the ORCID site and allow you to create a new iD or authenticate a pre-existing iD in Editorial Manager. Please see the following video for instructions on linking an ORCID iD to your Editorial Manager account: https://www.youtube.com/watch?v=_xcclfuvtxQ".

- The ORCID i.d of the corresponding author has been added.

3. In the online submission form, you indicated that [The data will be made available upon reasonable request from first author.]. 

 The de-identified data has been placed into the OSF repository and made public, available here DOI 10.17605/OSF.IO/4VXGE

Reviewer 1

My issue is that some of the crafted items, e.g., “Witnessed my dependents’ health suffer due to inadequate care from professional providers or interruptions in services,” “Witnessed my dependents’ education or overall development suffer due to interruptions in schooling/services” and “Worked with team members who do not treat those with COVID with dignity and respect” do not fit the standard definition of moral distress as coming from constraints on an individual’s own actions and I believe reflected in the authors’ (perhaps too vague) conceptualization of moral distress (lines 71-72), “Moral distress can be conceptualized broadly as challenges faced by those unable to provide care they believe to be required.” Restated, by the survey measurement items selected, the authors’ definition of moral distress includes one’s distress from witnessing others’ actions or outside events that one feels are wrong, and not just distress stemming from actions taken when having one’s own actions constrained. It also includes distress from actions that are constrained from happening, and not just constraints forcing actions.

This is a definitional challenge for the entire field of moral distress and not unique to this study. I believe the authors do not disagree. The field needs to recognize that even some of its accepted measurement instruments take the concepts of moral distress beyond what stems from constraints on one’s own actions. If the authors see the field this way, here is one definition that acknowledges that moral distress stems from more than solely what an individual is forced to do themselves: “we conceptualise moral distress as the psychological unease or distress that occurs when one witnesses, does things or fails to do things that contradict deeply held moral and ethical beliefs and expectations.” (Pathman et al. Moral distress among clinicians working in U.S. safety net practices during the COVID-19 pandemic: A Mixed Methods Study. BMJ Open. 2022;12:e061369. doi: 10.1136/bmjopen-2022-061369). Clear and accurate definitions of a study’s fundamental concepts are, of course, crucial in science.

- This is a very valuable comment, and we agree that the field of moral distress is characterized by definitional challenges. Thank you for pointing to a definition that does indeed reflect our approach in this instance. We have replaced the definition we included, which we recognize was rather vague, with the one provided. 

- In terms of the survey items, these were adapted from the Moral Distress Scale (MDS-R) which is a validated tool for measuring moral distress, which includes items such as ‘Watch patient care suffer . . ‘, which we adapted to ‘witness my dependent’ health suffer’. However, we do recognize that while such questions imply a distress from witnessing suffering and being constrained from taking action, they do not explicitly state this. This is something we will consider in future research.

Specific comments:

1. Lines 22-23 I find the phrase of the abstract obtuse: “little research considers the interaction between these two sectors of the care economy.” Perhaps this can be reworded for clarity for those minimally familiar with the ins and outs of unpaid caregiving.

- We have reworded this

2. Line 119. Please clarify/explain/reword “extractive responses”. This sounds like jargon and perhaps too value laden for scientific writing.

- There is a feminist literature on extractive care economies, but you are right that it is not clearly articulated in this paper and digging into that literature would probably be a distraction, so we have edited the sentence. 

3. Do the data allow investigators to know how both moral distress for both unpaid and paid care differ for subjects whose dependents include children? It would seem that the sources and amount of moral distress will differ when speaking of responsibilities and witnessing things happening to one’s own children versus other dependents. And given societal gender differences in responsibility for children (as the authors point out), how does parenthood affect moral distress differences in paid and unpaid care found for men and women?

- In order to keep the survey short we did not ask moral distress questions for each dependent type and so are unable to analyze results for those subjects whose dependents include children / are parents. This is a limitation we discussed during the analysis as something we would change in future and which I have now added to the limitation section here. 

4. Lines 350-362. The limitations listed are appropriate—I appreciate the investigators’ understanding of the limitations of their study and forthrightness in disclosing these limitations.

- Thank you!

5. How are study findings specific to or potentially affected by the timing of the survey, i.e., at about 20 months into the pandemic, when PPE shortages had abated, and an effective vaccine was widely available?

- We asked participants to reflect on their experience over the past year (between October 2021 and time of completing the survey which was October to December 2022). We have noted this in the methods section as well as in the limitations section

6. Conclusion section. I find this section emphasizes study findings related to gender differences and omits conclusions from study findings about the amount and relationship between moral distress from unpaid and paid care, which is the manuscript’s title and the study’s seeming focus within the abstract.

- We have added greater analysis about the relationship between unpaid and paid care and moral distress to the conclusion.

Reviewer #2: 

Given the depth of the analysis and the importance of the topic, I recommend a major revision to strengthen the manuscript’s impact and relevance. Enhancing the discussion section with clearer implications for practice, additional policy recommendations, and a more detailed exploration of the intersectionality of gender and other identity factors would be beneficial.

1- In method: Clarify the rationale behind the selection of specific items for the moral distress scales and how they align with the study objectives and previous research findings.

- have expanded on the process of selecting specific items for the moral distress scale

2- Emphasize the practical implications of the findings for healthcare organizations, policymakers, and support systems for healthcare workers.

- Added to conclusion

3- Address the study’s limitations, such as the non-random sampling and the exclusion of intersecting identity factors, and provide suggestions for future research directions to further explore these aspects.

- We have added this to methods section

4- Consider expanding on the recommendations for addressing gender norms, promoting men’s involvement in caregiving, and advocating for supportive care policies in healthcare settings.

- We have added content on this in the discussion and conclusion

5- In conclusion, with some revisions to enhance the practical implications and address study limitations, the manuscript has the potential to make a significant contribution to the field of healthcare ethics and gender studies. I look forward to seeing the revised manuscript and the valuable insights it will provide to the academic community.

- Thank you very much for the constructive feedback.

---

## [Decision Letter · Decision Letter 1]

26 Aug 2024

Moral distress related to paid and unpaid care  among healthcare workers  during the COVID-19 pandemic

PONE-D-24-17880R1

Dear Dr. Tiwana,

We’re pleased to inform you that your manuscript has been judged scientifically suitable for publication and will be formally accepted for publication once it meets all outstanding technical requirements.

Kind regards,

Alejandro Botero Carvajal, MD

Academic Editor

PLOS ONE

Additional Editor Comments (optional):

Reviewers' comments:

Reviewer's Responses to Questions

**Comments to the Author**

1. If the authors have adequately addressed your comments raised in a previous round of review and you feel that this manuscript is now acceptable for publication, you may indicate that here to bypass the “Comments to the Author” section, enter your conflict of interest statement in the “Confidential to Editor” section, and submit your "Accept" recommendation.

Reviewer #1: All comments have been addressed

Reviewer #2: All comments have been addressed

2. Is the manuscript technically sound, and do the data support the conclusions?

Reviewer #1: Yes

Reviewer #2: Yes

3. Has the statistical analysis been performed appropriately and rigorously? 

Reviewer #1: No

Reviewer #2: N/A

4. Have the authors made all data underlying the findings in their manuscript fully available?

Reviewer #1: (No Response)

Reviewer #2: Yes

5. Is the manuscript presented in an intelligible fashion and written in standard English?

Reviewer #1: Yes

Reviewer #2: Yes

6. Review Comments to the Author

Reviewer #1: I believe the authors have been responsive to all reviewers' questions and recommendations. I find the revised draft to be a solid paper that presents a solid study clearly of an understudied topic, with balanced tone.

Two suggested copy editing points:

Line 326--"their" is misspelled

Line 305--the use of the slash ("/") here is ambiguous (as it often is). I suggest writing out its intended meaning, which might be simply "limited" or "sparse."

Reviewer #2: The authors have effectively incorporated the feedback into the revised manuscript, enhancing the clarity, depth, and overall quality of the research presented.

7. PLOS authors have the option to publish the peer review history of their article (what does this mean?). If published, this will include your full peer review and any attached files.

Reviewer #1: **Yes: **Donald Pathman, MD MPH

Reviewer #2: **Yes: **Hadi Abbasian
